# Intra-Tumoral Angiogenesis Is Associated with Inflammation, Immune Reaction and Metastatic Recurrence in Breast Cancer

**DOI:** 10.3390/ijms21186708

**Published:** 2020-09-13

**Authors:** Masanori Oshi, Stephanie Newman, Yoshihisa Tokumaru, Li Yan, Ryusei Matsuyama, Itaru Endo, Masayuki Nagahashi, Kazuaki Takabe

**Affiliations:** 1Breast Surgery, Department of Surgical Oncology, Roswell Park Comprehensive Cancer Center, Buffalo, New York, NY 14263, USA; masa1101oshi@gmail.com (M.O.); snewman5@buffalo.edu (S.N.); Yoshihisa.Tokumaru@roswellpark.org (Y.T.); 2Department of Gastroenterological Surgery, Yokohama City University Graduate School of Medicine, Yokohama 236-0004, Japan; ryusei@yokohama-cu.ac.jp (R.M.); endoit@med.yokohama-cu.ac.jp (I.E.); 3Department of Surgery, Jacobs School of Medicine and Biomedical Sciences, State University of New York, Buffalo, New York, NY 14263, USA; 4Department of Surgical Oncology, Graduate School of Medicine, Gifu University, 1-1 Yanagido, Gifu 501-1194, Japan; 5Department of Biostatistics & Bioinformatics, Roswell Park Comprehensive Cancer Center, Buffalo, New York, NY 14263, USA; li.yan@roswellpark.org; 6Division of Digestive and General Surgery, Niigata University Graduate School of Medical and Dental Sciences, Niigata 951-8520, Japan; masanagahashi@gmail.com; 7Department of Breast Surgery, Fukushima Medical University School of Medicine, Fukushima 960-1295, Japan; 8Department of Breast Surgery and Oncology, Tokyo Medical University, Tokyo 160-8402, Japan

**Keywords:** angiogenesis, breast cancer, epithelial-mesenchymal transition, gene set, metastatic recurrence, sphingosine-1-phosphate

## Abstract

Angiogenesis is one of the hallmarks of cancer. We hypothesized that intra-tumoral angiogenesis correlates with inflammation and metastasis in breast cancer patients. To test this hypothesis, we generated an angiogenesis pathway score using gene set variation analysis and analyzed the tumor transcriptome of 3999 breast cancer patients from The Cancer Genome Atlas Breast Cancer (TCGA-BRCA), Molecular Taxonomy of Breast Cancer International Consortium (METABRIC), GSE20194, GSE25066, GSE32646, and GSE2034 cohorts. We found that the score correlated with expression of various angiogenesis-, vascular stability-, and sphingosine-1-phosphate (S1P)-related genes. Surprisingly, the angiogenesis score was not associated with breast cancer subtype, Nottingham pathological grade, clinical stage, response to neoadjuvant chemotherapy, or patient survival. However, a high score was associated with a low fraction of both favorable and unfavorable immune cell infiltrations except for dendritic cell and M2 macrophage, and with Leukocyte Fraction, Tumor Infiltrating Lymphocyte Regional Fraction and Lymphocyte Infiltration Signature scores. High-score tumors had significant enrichment for unfavorable inflammation-related gene sets (interleukin (IL)6, and tumor necrosis factor (TNF)α- and TGFβ-signaling), as well as metastasis-related gene sets (epithelial mesenchymal transition, and Hedgehog-, Notch-, and WNT-signaling). High score was significantly associated with metastatic recurrence particularly to brain and bone. In conclusion, using the angiogenesis pathway score, we found that intra-tumoral angiogenesis is associated with immune reaction, inflammation and metastasis-related pathways, and metastatic recurrence in breast cancer.

## 1. Introduction

Angiogenesis is defined as the creation of new blood vasculature from preexisting ones. It is considered one of the hallmarks of cancer [1] because blood vessels not only provide a conduit to supply oxygen and nutrients to cancer cells, but they also deliver molecules that confer immune resistance and facilitate the ability of cancer cells to metastasize [2,3].

A wide variety of genes are involved in angiogenesis. Vascular endothelial growth factors (VEGFs) and their receptors (VEGFRs) are the best studied angiogenic signaling pathways [4,5]. However, it is reported that several other gene pathways also play an important role in angiogenesis [6,7]. We have previously shown that two other pathways, Angiopoietin (ANGPT)- and Tyrosine kinase with immunoglobulin-like and EGF-like domains (Tie)-signaling, are associated with breast cancer survival [8]. Our group and others have demonstrated that a bioactive lipid mediator, sphingosine-1-phosphate (S1P), which is produced by sphingosine kinase 1 (SphK1) facilitates angiogenesis in breast cancer [9,10,11], and that SphK1 promotes breast cancer metastasis [12,13]. In breast cancer, metastasis is associated with increased inflammation and immune cell infiltration in tumors [14,15,16,17,18,19,20]. 

A multitude of conditions, including hypoxia and localized inflammation, coincide to promote tumor angiogenesis. Pro-inflammatory cytokines such as interleukin (IL)-1β, IL-1α and IL-6 work through a variety of mediators, including tumor necrosis factor (TNF)α, to either enhance or suppress angiogenesis [21,22,23] to a degree that is directly related to cytokine concentration and duration of exposure [24,25]. A combination of these factors may contribute to the tumor’s ability to invade vessels and metastasize to other parts of the body [26,27,28]. 

Despite these well-known mechanisms, the clinical relevance of targeting angiogenesis in breast cancer is somewhat ambiguous. Bevacizumab, an anti-VEGF antibody that targets tumor angiogenesis, was granted “accelerated” approval by the FDA in 2008 after results of the E2100 randomized phase 3 trial for the first-line treatment of human epidermal receptor 2 (HER2)-negative metastatic breast cancer. However, this decision was reversed in 2010 due to a lack of efficacy in response and survival [29].

Our group has demonstrated the efficacy of the gene set variation analysis (GSVA) method to summarize the expression of dozens of genes of a specific cellular pathway as a score in order to estimate the degree of the activation of that pathway. With this GSVA-based scoring system, we have shown the clinical relevance, such as prognosis and treatment response, for the KRAS [30], G2M checkpoint [31], and E2F [32] pathways in breast cancer. Utilizing this methodology, we developed the angiogenesis pathway score to quantify angiogenesis in a tumor using whole tumor transcriptomic data. Here, we hypothesize that intra-tumoral angiogenesis as measured with this score correlates with inflammation and metastasis in breast cancer patients.

## 2. Results

### 2.1. High Tumor Angiogenesis Pathway Score Is Associated with High Expression of Angiogenesis-, Hypoxia-, and Sphingosine-1-Phosphate (S1P)-Related Genes

The angiogenesis pathway score was determined from global tumor gene expression data as the GSVA score for the Molecular Signatures Database (MSigDb) Hallmark angiogenesis gene set. Genes of this gene set are listed in Appendix A. Using the median value of the score in a given cohort of patients as the cut-off, the patients of the cohort were binned into high and low score groups. In both The Cancer Genome Atlas (TCGA) [33] and Molecular Taxonomy of Breast Cancer International Consortium (METABRIC) [34] breast cancer cohorts, the median value separated the two weak peaks of score distributions (Appendix A). Note that the tumor samples obtained for these cohorts for gene expression measurement are representative of bulk tumors. Samples were confirmed to have high cancer cell content and no necrosis. To this end, we state our findings to be “intra-tumoral angiogenesis”. The common gold standard for evaluating angiogenesis is by immunostaining for determining micro-vessel density in peritumoral regions [35,36].

We first investigated whether expression of angiogenesis-related genes is elevated in tumors with a high angiogenesis score to determine if our score adequately captures intra-tumoral angiogenesis. VEGFs are well-studied angiogenetic factors. As shown in Figure 1A, the expression of multiple VEGF-related genes (*VEGFA, VEGFB, VEGFR1* (also known as *FLT1*), *VEGFR2* (*KDR*), and *VEGFR3* (*FLT4*), were significantly increased in high angiogenesis score tumors. However, the angiogenesis score had no association with expression of *VEGFA*, the only gene among those examined for Figure 1 that is in the Hallmark angiogenesis gene set (Appendix A). The pathway score thus significantly correlated with expression levels of four of the five angiogenesis-related genes, suggesting that the score adequately delineates intra-tumoral angiogenesis. Our angiogenesis score calculation considers the expression of 35 other genes besides *VEGFA*. This allows quantification of a complicated biological process such as angiogenesis because it is known that expressions of each gene intertwine with each other during these processes and measurements of a single gene most likely are not enough to grasp the whole picture. It therefore is possible, as seen in Figure 1, for expression of *VEGFA* to not correlate with the angiogenesis score.

Endothelial cell surface marker genes, *CD31* (*PECAM1*) and von Willebrand factor (*VWF*), were also significantly elevated in high score tumors (both *p* < 0.001) indicating that high score is associated with increased concentration of blood vessels inside a tumor (Figure 1B). 

All the genes that we have previously reported to represent vascular stability [8,37,38], *TIE1*, *TIE2* (*TEK*), Angiopoietin (*ANGPT*) 1 and 2, Vascular endothelial cadherin (*CDH5*), Claudin 5 (*CLDN5*), and Junction adhesion molecule 2 (*JAM2*), demonstrated a significant increase in high score tumors (all *p* < 0.001, Figure 1C). The hypoxia-related gene, Hypoxia-inducible factor 1-alpha (*HIF1A*) was elevated in high score tumors in both cohorts. Another hypoxia-related gene, HIF1-beta (*HIF1B*, or *ARNT*) was increased in high score tumors in the TCGA cohort, but not in the METABRIC cohort (Figure 1D). By the nature of our study, the causality of whether angiogenesis is the cause of hypoxia or vice versa is unknown. While hypoxia regulates *HIF1A/HIF1B* activity at the post-transcription level, it also increases *HIF1A/HIF1B* gene expression [39]. This highlights the caveat of using the score, that although it does demonstrate association, the causality remains unknown.

We and others have repeatedly demonstrated that S1P plays a critical role in angiogenesis [10,11,12,17,19,20,40,41,42]. In agreement with our mechanistic model [12], elevation of *SphK1*, suppression of sphingosine kinase 2 (*SphK2*), elevation of *SPNS2* and Sphingosine-1-phosphate kinase receptor-1 (*S1PR1*), all of which work to export S1P and promote angiogenesis, were associated with high score tumors (all *p* < 0.001, Figure 1E). These results were mirrored in both TCGA and METABRIC cohorts (*SPNS2* expression data is unavailable for METABRIC). Furthermore, we investigated the association of the angiogenesis score with expression of tumor suppressor genes (Tumor p53 (*TP53*), Phosphatase and tensin homolog (*PTEN*), von Hippel-Lindau tumor suppressor (*VHL*), CD95 [*FAS*], and Suppression of tumorigenicity 5 (*ST5*)) as well as *NOTCH1* and *NOTCH4* genes, which have been reported to associate with vasculogenesis [43]. *PTEN, CD95, ST5,* and *NOTCH1* and *NOTCH4* genes were significantly elevated in high score tumors in both cohorts (*NOTCH4* expression data is unavailable for METABRIC). The angiogenesis score had no association with *TP53* expression in either cohort, and its associations with *VHL* were opposite in the TCGA and METABRIC cohorts (Appendix A). 

### 2.2. Tumor Angiogenesis Score Is not Associated with Aggressive Clinical Features or Response to Neoadjuvant Chemotherapy (NAC)

Since angiogenesis is one of the hallmarks of cancer, we hypothesized the amount of intra-tumoral angiogenesis to be associated with clinical aggressiveness. As expected, the angiogenesis score was significantly associated with Nottingham pathological grade and American Joint Committee on Cancer (AJCC) cancer staging in METABRIC (*p* = 0.009 and *p* < 0.001) but not in TCGA cohort, although tumors with metastasis (Stage IV, N3, M1) demonstrated a notable trend to be higher score (Figure 2A). The score was not associated with subtypes in either of the cohorts.

In the past, there was a notion that a high density of angiogenesis in a tumor is a good predictive indicator of treatment efficacy since improved microcirculation facilitates drug delivery to cancer cells [44]. However, it is now believed that angiogenesis in tumor can be pathologic and dysfunctional and does not improve microcirculation or drug delivery [45]. This aligns with our observation that the angiogenesis score does not negatively correlate with hypoxia (Figure 1D). To this end, it was of interest to examine whether the angiogenesis score is associated with response to NAC. For this, we utilized clinical and tumor gene expression data from three independent cohorts, with Gene Expression Omnibus repository identifiers GSE21094 (treated with paclitaxel/5-fluorouracil/cyclophosphamide and doxorubicin) [46], GSE25066 (taxane and anthracycline) [47], and GSE32646 (5/fluorouracil/epirubicin/cyclophosphamide and paclitaxel) [48]. As expected, the pre-NAC tumor angiogenesis score was not associated with the rate of pathological complete response (pCR) in patients with either estrogen receptor (ER)-positive/human epidermal receptor 2 (HER2)-negative or triple negative breast cancer (TNBC) tumors (Figure 2B).

### 2.3. Tumor Angiogenesis Score Is not Associated with Survival

Given the role of angiogenesis in tumor progression, we expected that patients with tumors of high angiogenesis scores will have a worse disease outcome. Contrary to this expectation, there was no association between the angiogenesis score and outcome measured as progression-free survival (PFS), disease-free survival (DFS), disease specific survival (DSS), or overall survival (OS), in either the TCGA or METABRIC cohort for all or specific breast cancer subtypes (all log-rank test *p* > 0.05; Figure 3 and Appendix A).

### 2.4. Tumors with High Angiogenesis Score Have Less Infiltration of Both Favorable and Unfavorable Immune Cells

Based on the notion that enhanced angiogenesis improves microcirculation of a bulk tumor, we expected that tumor infiltration by immune cells will be higher in high angiogenesis score tumors. To examine this, we estimated the abundance of different subsets of infiltrating immune cells using the xCell algorithm [49], and found that high angiogenesis score tumors had significantly less infiltration of multiple immune cells, including both favorable: CD4 memory, T helper 1 (Th1) and B cell (Figure 4A), and unfavorable immune cells: T helper 2 (Th2) and Regulatory T cell (Treg) in both TCGA and METABRIC cohorts (Figure 4B). On the other hand, Dendritic cell (DC) and M2 macrophage were highly infiltrated in high angiogenesis score tumors in both cohorts (Figure. 4A, B). Additionally, high score tumors were significantly associated with lower infiltration of CD8 T cells in the METABRIC, but not TCGA cohorts (Figure. 4A). In order to further elucidate the relationship between the angiogenesis score and cancer immunity, we further analyzed the association of the angiogenesis score with several other scores: Leukocyte Fraction, Lymphocyte Infiltration Signature Score and Tumor infiltrating lymphocyte (TIL) Regulation scores, using previously reported datasets [50] (Figure 4C). TIL regional fraction was higher in the low angiogenesis score group, while the Leukocyte fraction and lymphocyte infiltration signature showed higher values in the angiogenesis score high tumors (Figure. 4C). These results demonstrate that there is no simple relationship between intra-tumoral angiogenesis and infiltration of immune cells, nor with favorable or unfavorable immune cells.

### 2.5. Tumors with High Angiogenesis Score Have Significantly Enriched Expression of Multiple Inflammation-Related Gene Sets

To further understand why there was no evident survival difference in high and low angiogenesis score groups, we performed Gene set enrichment analysis (GSEA) of Hallmark gene sets in both TCGA and METABRIC cohorts. As expected, high angiogenesis score tumors demonstrated significantly enriched immune response-related Hallmark gene sets; interferon (IFN)-γ response, and IL2-STAT5 signaling (Figure 5A). Interestingly, high angiogenesis score tumors also contained significantly enriched unfavorable inflammation-related Hallmark gene sets, inflammatory response, IL6-JAK-STAT3 signaling, TNF-α signaling via NFkB, and TGF-β signaling and hypoxia. (Figure 5B). Furthermore, we found that metastasis-related gene sets; including epithelial mesenchymal transition (EMT), HEDGEHOG signaling, NOTCH signaling and WNT-β catenin signaling were also enriched in angiogenesis score high tumors (Figure 5C). In particular, EMT pathway score was strongly correlated with angiogenesis pathway score in both the TCGA and METABRIC cohorts (Figure 5D; spearman r = 0.894 (*p* < 0.01) and r = 0.868 (*p* < 0.01), respectively). These results suggest that an intra-tumoral angiogenesis high tumor is associated with favorable immune response, but also with unfavorable inflammation, hypoxia, and correlated with metastasis-related signaling, particularly EMT.

### 2.6. Tumor with High Angiogenesis Scores Were Significantly Associated with Metastatic Recurrence

Given the above results, we further analyzed the association of the angiogenesis score in the primary tumor with the time it took to develop metastasis at specific sites using the GSE2034 cohort. Kaplan–Meier analyses of site-specific metastasis-free survival demonstrated that the score was associated with brain and bone, but not with lung metastatic recurrence in this cohort (Figure 6). These results suggest that intra-tumoral angiogenesis of the primary tumor is associated with metastatic recurrence to the brain or bone.

## 3. Discussion

Here, we studied the clinical relevance of the intra-tumoral angiogenesis using the angiogenesis pathway score, which is determined by the GSVA score of the Hallmark angiogenesis gene set on multiple large patient cohorts with transcriptome data. High angiogenesis pathway score tumor was significantly associated with the expression of multiple genes including, VEGF-, vascular stability-, Hypoxia-, and S1P-repated genes. The high score was not associated with clinical features or survival outcome. Angiogenesis pathway score high-tumor was significantly associated with low fractions of both favorable and unfavorable immune cells, except for dendritic cells and M2 macrophages. Furthermore, angiogenesis pathway score high-tumors were also associated with other immune-related scores, including leukocyte fractions, TIL regional fractions and lymphocyte infiltration signature. Angiogenesis score high-tumor significantly enriched not only immune-related pathways, but also inflammation-related pathways such as inflammatory response, IL6, TNFα signaling, TGFβ signaling, as well as metastasis-related gene pathways including EMT, and Hedgehog-, notch- and WNT-β catenin-signaling. Interestingly, angiogenesis score high-tumors were significantly associated with metastatic recurrence especially to the brain and bone.

We utilized a angiogenesis pathway score that analyzed 36 gene expressions to grasp the complex mechanism of angiogenesis. Using this approach, our lab has previously demonstrated that KRAS signaling score high-triple negative breast cancer is associated with favorable tumor immune microenvironment and better survival [30], that G2M cell cycle pathway score is a prognostic biomarker of metastasis in ER-positive breast cancer [31], and that E2F pathway score is a predictive biomarker of response to neoadjuvant therapy in ER+/HER2− breast cancer [32]. Previously we found that expression of angiopoietin signaling, but not VEGF, was associated with breast cancer survival, which is in agreement with a notion that expression of a single factor may mislead the big picture [8]. Indeed, angiogenesis pathway score correlated with the 18 angiogenesis related gene expressions we investigated except for VEGF-A and HIF1B genes.

Our group was the first to demonstrate the mechanism of how S1P is generated inside a breast cancer cell by SphK1, where SphK2 is often suppressed in a compensatory manner, and is then exported out of breast cancer cells by transporters [51] including SPNS2 [52]. S1P exported out of the cell binds to S1P receptors, known as inside-out signaling of S1P [12,40], and plays an important role in angiogenesis and vascular stability [11]. In agreement with our mechanistic model, we showed that human breast cancers with a high amount of angiogenesis are significantly associated with high expression of S1P-signaling genes, SphK1, S1PR1, and SPNS2.

In the current study, we found that intra-tumoral angiogenesis high-patient tumors had a significantly enriched inflammatory response, including IL6 signaling, TNFα signaling, and TGFβ signaling gene sets. This is strikingly consistent with our mechanistic model that S1P is persistent following STAT3 activation, chronic inflammation and development and progression of colon cancer [53] as well as breast cancer [53]. Furthermore, our results showing that hypoxia was enriched in angiogenesis high-tumors is consistent with our in vitro finding that S1P regulates hypoxia-inducible factor functions [54].

Contrary to our expectations, however, the angiogenesis score was not associated with aggressive clinical features, such as subtype, clinical staging, pathological grade, nor with survival outcome. This may at least partially be explained by our findings that both favorable and unfavorable tumor immune microenvironments were mixed in angiogenesis high tumors. Tumors with high angiogenesis scores were associated with smaller fractions of not only favorable but also unfavorable immune cells. We further found the enrichment of favorable immune-related, as well as unfavorable inflammation-related Hallmark gene sets in the high angiogenesis score tumors. These results may suggest that the clinical impact of angiogenesis depends on the balance of favorable and unfavorable reactions.

In the current study, we found that metastasis-related gene sets, including EMT, Hedgehog-signaling, Notch-signaling and WNT-signaling gene sets, were significantly enriched in tumors with high angiogenesis score. Furthermore, it was significantly associated with worse site-specific metastasis-free survival, especially in the brain and bone. We have previously reported that SphK1-generated S1P links chronic inflammation and metastasis [13,17,55,56], and shortens the survival of mice and patients with breast cancer [10,41]. Given our mechanistic work on the role of S1P signaling in breast cancer and our findings of clinical relevance to angiogenesis in breast cancer in the current study, we cannot help but speculate that S1P may play a critical role in this process. However, in order to prove this speculation, further data on patient breast cancer, such as direct measurement of S1P in tumors by mass spectrometry and evaluation of S1P productivity in tumors, is necessary.

We have utilized the gene set variation analysis (GSVA) score of the Molecular Signatures Database (MSigDb) Hallmark angiogenesis gene set as the angiogenesis score in our study. The Hallmark angiogenesis gene set was generated by MSigDb [57] and has been used in hundreds of studies to date. This score constitutes of 36 genes that allow quantification of complicated multi-gene pathways, such as angiogenesis where there is significant cooperation and a single gene is insufficient to fully encapsulate the biological processes at play. In alignment with this notion, we found that the angiogenesis score associates with other known angiogenesis-associated gene expressions except for VEGF-A and HIF1B genes. Further, the intra-tumoral angiogenesis score was associated with immune reaction-, inflammation-, and metastasis-related pathways, and metastatic recurrence in breast cancer. This is in alignment with our general expectation. However, the score did not correlate with survival, whereas pathologically determined angiogenesis is known to associate with breast cancer outcome [58,59]. One of the explanations for this difference may be the spatial location of whether angiogenesis is evaluated in intra- or peri-tumoral region of the bulk tumor. In TCGA, tumor samples used for transcriptome profiling were pathologically confirmed to have more than 60% of cells as cancer cells with no necrosis [60] which indicates a viable center of the bulk tumor, whereas angiogenesis is commonly evaluated at the edge of the bulk tumor pathologically by microvessel density as the gold standard. The novel finding in this paper is that the angiogenesis score, although it is not a gold standard, allows assessment of angiogenesis in any sample with transcriptomic data.

The angiogenesis pathway score that measures patients’ tumor angiogenesis could become a useful tool to decide management in the future; however, this study is not free from limitations. First of all, it needs to be noted that the difference in the location in the tumor where the sample is taken from may significantly influence the result. Thus, we state that our finding is in regard to intra-tumoral angiogenesis. Furthermore, our analyses are limited to measurement of gene expression whereas the gold standard to evaluate angiogenesis is micro-vessel density, which may also contribute to the difference in results. Although we have validated the score using multiple largest publicly available breast cancer cohorts, the angiogenesis pathway score should be used in a prospective cohort to fully explore its utility.

## 4. Materials and Methods

### 4.1. Genetic Profiling and Clinical Information of Breast Cancer Cohorts

The Cancer Genome Atlas Breast Cancer cohort (TCGA-BRCA) RNA-sequencing data and clinical information of 1065 patients, who were female and had pathological diagnosis of breast cancer, were obtained from Pan-Cancer Clinical Data Resource [33] and through the cBio Cancer Genomic Portal [61]. The data of 1903 cases in the Molecular Taxonomy of Breast Cancer International Consortium (METABRIC) cohort [34] were also obtained from the cBioPortal. We also used normalized tumoral genomic and clinical data provided by the Gene Expression Omnibus (GEO) repository of the US National Institutes of Health (http://www.ncbi.nlm.nih.gov/geo, accessed on 20 June 2020). For genes with multiple probes, the average value was used. Gene expression data were transformed for log_2_ in all analyses. We used the studies of Shi et al. (GSE20194; *n* = 197) [46], Symmans et al. (GSE25066; *n* = 467) [47], Miyake et al. (GSE32646; *n* = 81) [48] and Wang et al. (GSE2034; *n* = 286) [62].

### 4.2. Gene Set Expression Analyses

Gene set variation analysis (GSVA) score of the “HALLMARK_ANGIOGENESIS” gene set of the MSigDb Hallmark [57] collection was used to measure the angiogenesis pathway score using the GSVA Bioconductor package (version 3.10) [63], similar to how we measured KRAS signaling score [30], G2M cell cycle pathway score [31] and E2F pathway score [32]. A false discovery rate (FDR) of less than 0.25 was used for the statistical significance in the GSEA analysis, which is recommended by GSEA software (Lava version 4.0) [64] as we previously reported [16,37,65,66,67,68,69,70,71,72].

### 4.3. Statistical Analysis

The median value of the angiogenesis pathway score was used to divide low and high angiogenesis score groups in each cohort. Analysis of comparisons between groups used one-way analysis of variance (ANOVA) and Fisher’s exact tests. Kaplan–Meier curves with log-rank tests were used for survival analyses. The xCell algorithm (version 16 for Windows; Microsoft, Redmond, WA, USA) was used to estimate the cell fraction of a tumor from its mRNA gene expression data [49]. *p* < 0.05 was used as statistically significant for all tests. Tukey type boxplots show median and interquartile level values. R software (version 4.0.1, R Project for Statistical Computing) and Excel (version 16 for Windows, Redmond, WA, USA) were used for mRNA data analysis and figures construction.

## 5. Conclusions

In conclusion, we developed an angiogenesis pathway score that is validated with many other angiogenesis-related genes. Using the score, we found that intra-tumoral angiogenesis is associated with immune reaction, inflammation and metastasis-related pathways and metastatic recurrence in breast cancer.

## Figures and Tables

**Figure 1 ijms-21-06708-f001:**
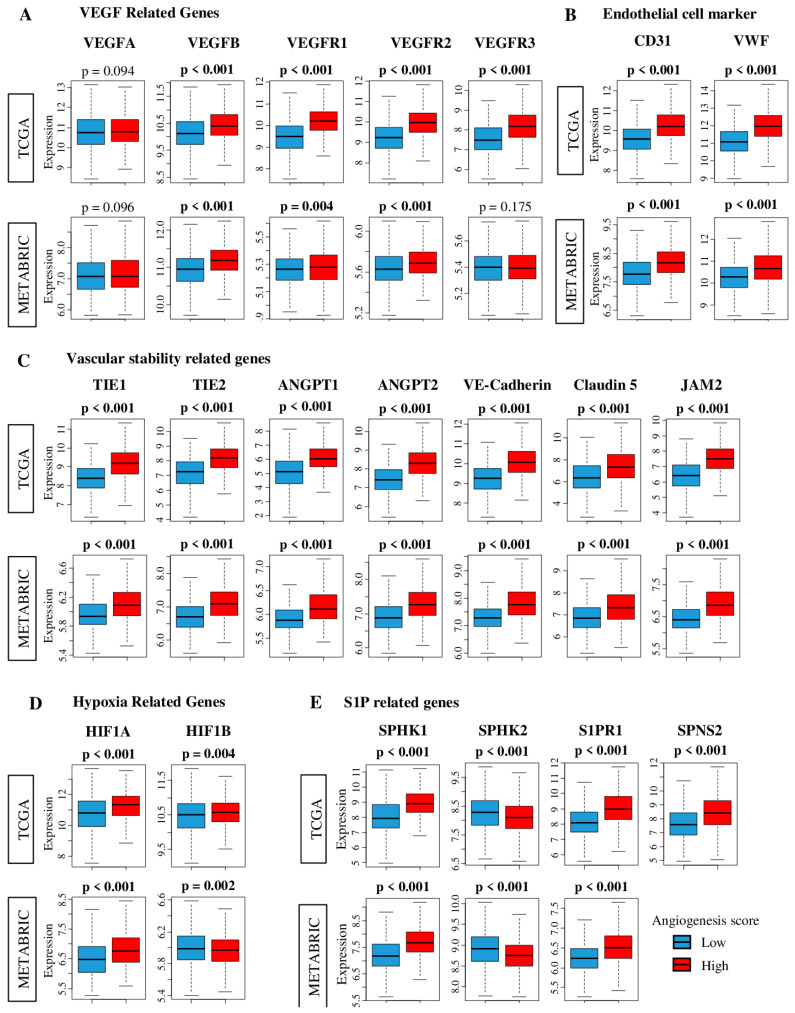
Association between tumor angiogenesis score and expression of Vascular endothelial growth factor (*VEGF*)-, blood endothelial cell markers-, vascular stability-, hypoxia-, and Sphingosine-1-Phosphate (S1P)-related genes in The Cancer Genome Atlas (TCGA) and Molecular Taxonomy of Breast Cancer International Consortium (METABRIC) cohorts. Red and blue boxes stand for high and low angiogenesis pathway score groups, respectively. Tukey type boxplots show median and inter-quartile level values. (**A**) Gene expression levels of *VEGF*-related genes; *VEGFA*, *VEGFB*, *VEGFR1 (FLT1*), *VEGFR2 (KDR*) and *VEGRR3 (FLT4*). (**B**) Gene expression levels of endothelial cell marker-related genes; *CD31 (PECAM1*) and von-Willebrand factor (*VWF)*. (**C**) Gene expression levels of vascular stability-related genes; *TIE1*, *TIE2*, *ANGPT1*, *AMGPT2*, VE-cadherin, Claudin5 and *JAM2*. (**D**) Gene expression levels of hypoxia-related genes; *HIF1A and HIF1B*. (**E**) Gene expression levels of S1P-related genes; *SphK1*, *SphK2*, *S1PR1*, and *SPNS2*. One-way ANOVA test was used to calculate *p* values. *ANGPT1/2*; Angiopoietin 1/2, *CD31* (gene name: *PECAM1*), Claudin 5 (gene name: *CLDN5*), Hypoxia-inducible factor 1-alpha (*HIF1A*), *HIF1B* (gene name: *ARNT*), *SPHK1/2*; Sphingosine kinase 1/2, *SPNS2*; Spinster homolog 2, *S1PR1*; Sphingosine-1-phosphate kinase receptor-1, *TEKVE*-cadherin; *TIE1*; Tyrosine kinase with immunoglobulin-like and EGF-like domains 1, Vascular endothelial cadherin (gene name: *CDH5*), Junction Adhesion Molecule 2 (*JAM2*), *VWF*; von Willebrand factor (*VWF*), *VEGFR1* (gene name: *FLT1*), *VEGFR2* (gene name: *KDR*), *VEGFR3* (gene name: *FLT4*), *VEGF*; Vascular endothelial growth factor.

**Figure 2 ijms-21-06708-f002:**
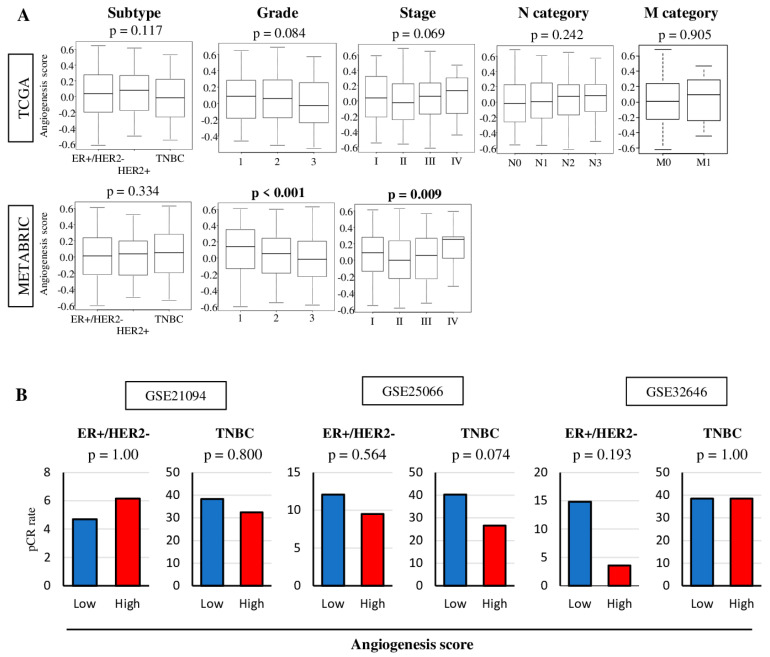
Association between clinical features and the tumor angiogenesis score in TCGA and METABRIC breast cancer cohorts. The median was used as a cut-off to divide patients into high and low score groups within each cohort. (**A**) Boxplots of the angiogenesis score by breast cancer subtype, Nottingham pathological grade, and American Joint Committee on Cancer (AJCC) cancer stage in the TCGA and METABRIC cohort, and AJCC N and M category in TCGA cohort. One-way ANOVA test was used to calculate *p* values. Tukey type boxplots show median and inter-quartile level values. (**B**) Pathological complete response (pCR) rate for neoadjuvant chemotherapy between low (blue) and high (red) angiogenesis pathway score in estrogen receptor-positive/human epidermal growth factor receptor 2-negative (ER+/HER2-) and triple negative breast cancer (TNBC) in the GSE20194 (*n* = 197), GSE25066 (*n* = 467), and GSE32646 (*n* = 81) cohorts. Two tailed fisher’s exact test was used to calculate *p* values.

**Figure 3 ijms-21-06708-f003:**
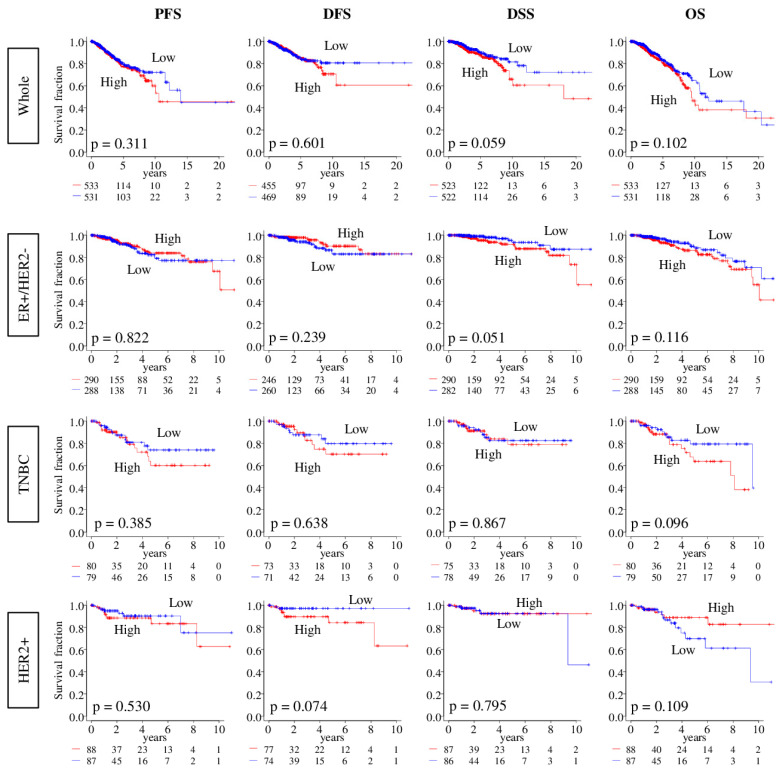
Association between tumor angiogenesis score and survival of breast cancer patients. Progression-free survival (PFS), Disease-Free (DFS), Disease-Specific (DSS), and Overall Survival (OS) of angiogenesis score low (blue) and high (red) in Whole breast cancer cohort, and each subtypes; estrogen receptor-positive/human epidermal growth factor receptor 2-negative (ER+/HER2-), triple negative breast cancer (TNBC), and HER2-positive, in the TCGA cohorts. The median was used as cut-off to divide into high and low score groups within each cohort. Log rank test was used to compare between two groups with Kaplan–Meier survival curves and to calculate *p* values.

**Figure 4 ijms-21-06708-f004:**
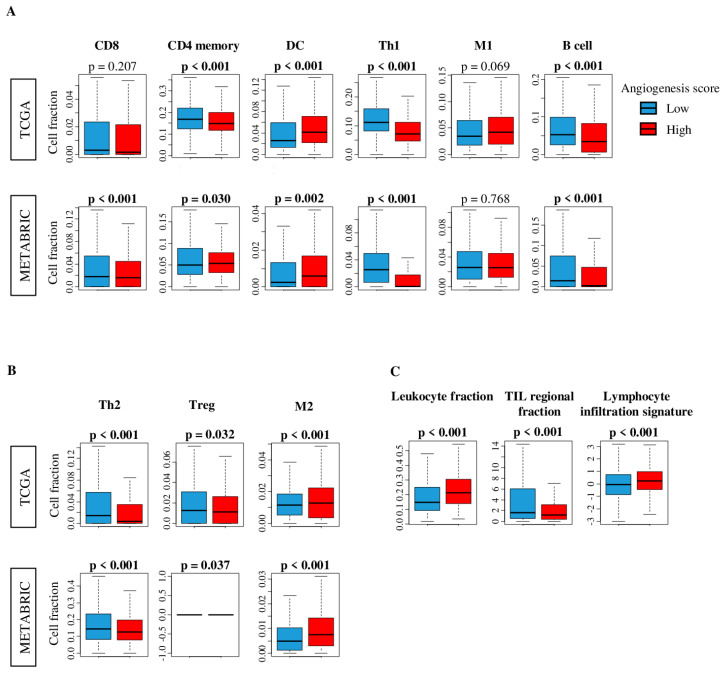
Comparison of tumor infiltrating immune cells between low (blue) and high (red) angiogenesis score tumors. Boxplots of the comparison with (**A**) favorable immune cells: CD8, CD4 memory, dendritic cell (DC), T helper type 1 cells (Th1), M1 macrophages, and B cell, and (**B**) unfavorable immune cells: T helper type 2 cells (Th2)*,* regulatory T cell (Treg), and M2 macrophage by low and high angiogenesis scores in the TCGA and METABRIC cohorts. (**C**) Comparison of low and high angiogenesis scores in scores of leukocyte fraction, tumor infiltrating lymphocyte (TIL) regional fraction and lymphocyte infiltration of TCGA cohort. The median was used as cut-off to divide into high and low score groups within each cohort. One-way ANOVA test was used to calculate *p* values. Tukey type boxplots show median and inter-quartile level values.

**Figure 5 ijms-21-06708-f005:**
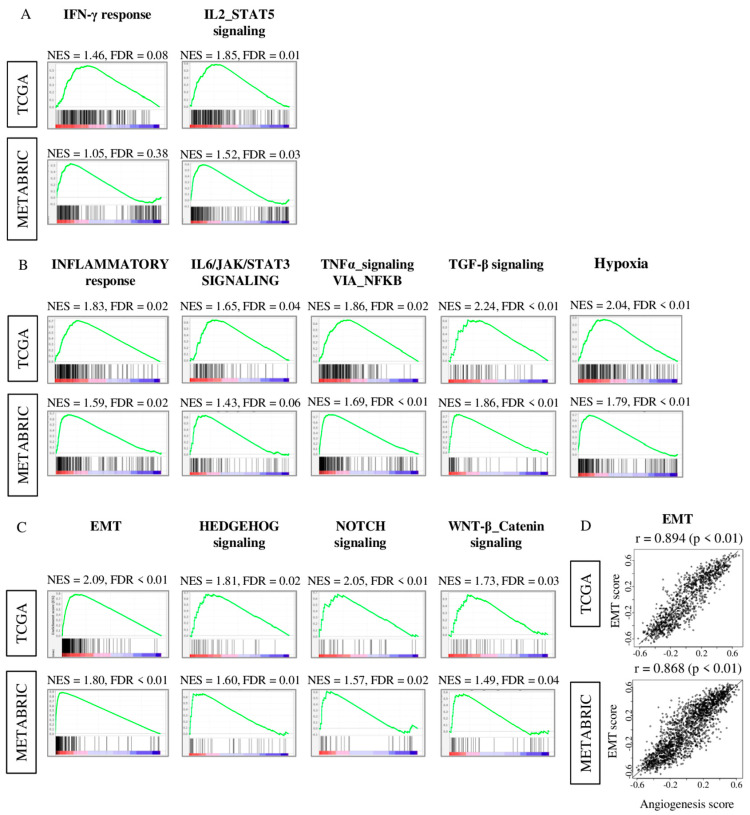
Gene Set Enrichment Assay (GSEA) comparing low and high angiogenesis pathway score tumors with enrichment gene sets in high angiogenesis pathway score group in both the TCGA and METABRIC cohorts. Correlation plot of (**A**) immune response gene sets; interferon (IFN)-γ, and IL2-STAT5 signaling, (**B**) Inflammatory response gene sets; Inflammatory response, IL6-JAK-STAT3 signaling, TNF-α signaling via NFkB, TGF-β signaling and hypoxia, (**C**) metastasis-related gene sets; epithelial mesenchymal transition (EMT), HEDGEHOG signaling, NOTCH signaling and WNT-β catenin signaling with normalized enrichment score (NES) and false discovery rate (FDR). The median was used as cut-off to divide into high and low score groups within each cohort. (**D**) Correlation plots of the angiogenesis score and EMT pathway score of both cohorts. *p* value was analyzed with spearman r correlation.

**Figure 6 ijms-21-06708-f006:**
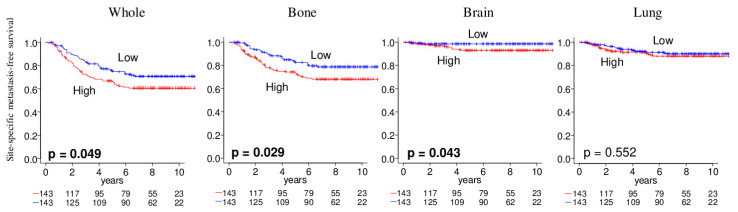
Association of the angiogenesis pathway score with metastatic recurrence in breast cancer. The Kaplan–Meier survival plots of metastasis-free survival for metastasis to bone, brain, or lung based on the pre-metastasis primary tumor in GSE2034 cohort (*n* = 286). The median was used as cut-off to divide into high (red) and low (blue) score groups within each cohort. To calculate *p* values, log rank test is used for comparing between two groups with Kaplan–Meier survival curves.

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
