# Peer review of "Intra-Tumoral Angiogenesis Is Associated with Inflammation, Immune Reaction and Metastatic Recurrence in Breast Cancer"

_ijms, 2020, doi:10.3390/ijms21186708_

Round 1

Reviewer 1 Report

Please correct the numerous  grammar mistakes starting  with the abstract and the  fact that in english the verbs take an s at the third person  in the present form.

sentences like : 
"sphingosine kinase 1 (SphK1) promotes angiogenesis in breast cancer [6,7] and its metastasis [8,9], which involves inflammation [10,11] and immune cells [12-14]." 

are not understandable.  It means that angiogenesis is making metatasEs,(that is very new!) and that metastases involve  inf.... 1 please put all that in the proper order.

Line  58 : "Hypoxia and inflammation are known mechanisms" NO hypoxia is not a mechanism it is a condition, this cannot be classified as inflammation which  results of a succession of mechanisms

line 60:" (TNF)α via NFkB  promote   ?????

Line  62: "thought to progress a tumor to vessel invasion and metastasis" this does not mean ANYTHING in English nor in any other language.

Author Response

Reviewer #1:

Comment 1:

Please correct the numerous  grammar mistakes starting  with the abstract and the  fact that in english the verbs take an s at the third person  in the present form.

Response 1:

First of all, we would like to thank Reviewer #1 for taking time and effort reviewing our manuscript. We apologize that our use of the English language was not of publication quality. We have carefully revised and corrected the grammar throughout the entire manuscript and marked the changes using “Track Changes” function.

Comment 2:

sentences like: "sphingosine kinase 1 (SphK1) promotes angiogenesis in breast cancer [6,7] and its metastasis [8,9], which involves inflammation [10,11] and immune cells [12-14]." are not understandable.  It means that angiogenesis is making metatasEs,(that is very new!) and that metastases involve  inf.... 1 please put all that in the proper order.

Response 2:

We totally agree with the Reviewer #1 that the description was confusing. We have re-written this sentence for clarity as below,

Introduction section:

“Our group and others have demonstrated that a bioactive lipid mediator, sphingosine-1-phosphate (S1P) produced by sphingosine kinase 1 (SphK1) facilitates angiogenesis in breast cancer [10-12], and that SphK1 promotes breast cancer metastasis [13,14]. In breast cancer, metastasis is associated with increased inflammation and immune cell infiltration in tumors [15-21].”

Comment 3:

Line 58: "Hypoxia and inflammation are known mechanisms" NO hypoxia is not a mechanism it is a condition, this cannot be classified as inflammation which  results of a succession of mechanisms

Response 3:

We agree with the Reviewer that hypoxia is not a mechanism and our original writing was incorrect. We have re-written the sentence as below,

Introduction section:

“A multitude of conditions, including hypoxia and localized inflammation, are thought to coincide to promote tumor angiogenesis.

Comments 4:

line 60:" (TNF)α via NFkB  promote   ?????

Line 62: "thought to progress a tumor to vessel invasion and metastasis" this does not mean ANYTHING in English nor in any other language.

Response 4:

We agree with the Reviewer that these sentences are unclear and inaccurate. We have now revised them as below,

Introduction section:

“Pro-inflammatory cytokines such as interleukin (IL)-1β, IL-1α and IL-6 work through a variety of mediators including tumor necrosis factor (TNF)α that could modulate both pro- and anti-angiogenic properties [22-24], which is directly related to the concentration of TNFα and duration of exposure [25,26].

Introduction section:

“A combination of these factors may contribute to the tumor’s ability to invade vessels and metastasis to other areas of the body [28-30].

Reviewer 2 Report

This manuscript used patient cohort data to correlate clinical with gene expression data.

Using the angiogenesis score (a set of gene expression) the authors show similarities to known angiogenesis related genes namely the VEGF group, yet get non-expected results for clinical outcomes and immune infiltration. Only the correlation of angiogenesis score to metastasis meet the general expectations of high angiogenesis beeing favorable for tumor growth and metastasis. In addition the authors state (and list in Table S1) that only on gene oft he VEGF group – and the only gene not reacting in the ‚correct‘ direction is included in the angiogenesis score set. This needs to be explained and discussed in much more detail, since it seems to this reviewer, that the genes et for angiogenesis score needs improvement. In addition, while gene expression analysis is quite practicable, angiogenesis related factors, e. g. HIF1alpha are mainly regulated on posttranscriptional level (including localization). It seems especially intreguing that high angiogenesis score tumors should be more hypoxic. The authors should comment on this in regard to the general usefullness vs caveats of the angiogenesis score. This reviewer agrees with the authors, that a main point of tumor assessment is the site within the tumor where samples where taken. Especially ‚inside‘ the tumor necrosis could be expected. The authors should comment on this – maybe this was corrected for.

The reference list should be more balanced, since more than half of the citations seem to come from the authors groups – here condensation would allow for citing pioneers of angiogenesis and of the key molecules involved.

Minor points:

Reference 4 does not define maximum tumor size without angiogenesis (it only states it).

Reference 24 is mainly lymphangiogenesis and does not refer to peritumoral localization of angiogenesis (measurements).

In Figure D the graph for HIF1B does not seem to display a higher value in median, yet the boxes of the plot are more condensed. This reviewer asked the authors to clarifiy ths impression of data presentation.

Are lines 125-131 on page 148 part of Figure Legend 1?

Some spelling and layout issues should be corrected.

Author Response

Reviewer #2:

Comment 1:

Using the angiogenesis score (a set of gene expression) the authors show similarities to known angiogenesis related genes namely the VEGF group, yet get non-expected results for clinical outcomes and immune infiltration. Only the correlation of angiogenesis score to metastasis meet the general expectations of high angiogenesis being favorable for tumor growth and metastasis. In addition the authors state (and list in Table S1) that only on gene of the VEGF group – and the only gene not reacting in the ‚correct‘ direction is included in the angiogenesis score set. This needs to be explained and discussed in much more detail, since it seems to this reviewer, that the genes set for angiogenesis score needs improvement.

Response 1:

We would like to thank Reviewer #2 for taking time and effort reviewing our manuscript and providing constructive criticism. We agree with the Reviewer that additional explanation and discussion on the gene set as well as our findings will strengthen our manuscript.

We have utilized the gene set variation analysis (GSVA) score of the Molecular Signatures Database (MSigDb) Hallmark angiogenesis gene set as the angiogenesis score in our study. The Hallmark angiogenesis gene set was generated by MSigDb (Liberzon et al., Cell, 2015) and have been used in hundreds of studies. This score constitutes of 36 genes that allows quantification of a complicated biological process such as angiogenesis, because it is known that expressions of each gene intertwine with each other during these processes and measurements of a single gene most likely is not enough to grasp the whole picture. In alignment with this notion, we found that the angiogenesis score associates with other known angiogenesis-associated gene expressions except for VEGFA, which happened to be the only gene that was included in the score among the ones we analyzed.

As the Reviewer pointed out, the intratumoral angiogenesis score was associated with immune reaction, inflammation and metastasis-related pathways and metastatic recurrence in breast cancer, which was in alignment with general expectation, although it did not correlate with survival. One of the explanations of this difference between our result and earlier reports may due to the difference in the spacial location of where angiogenesis is evaluated in the bulk tumor, intra- or peri-tumoral region. TCGA states that the samples corrected were required to and pathologically confirmed to have more than 60% of cancer cells included in the specimen, whereas gold standard to assess angiogenesis is by microvessel density at the edge of a bulk tumor. The novel finding in this paper is that angiogenesis score, although it is not a gold standard to measure the tumor angiogenesis, allow assessment of angiogenesis in any sample with transcriptomic data that also allow analyses of large number of samples. We have added these explanations in Discussion section as below.

Discussion section:

“We have utilized the gene set variation analysis (GSVA) score of the Molecular Signatures Database (MSigDb) Hallmark angiogenesis gene set as the angiogenesis score in our study. The Hallmark angiogenesis gene set was generated by MSigDb [49] and have been used in hundreds of studies to date. This score constitutes of 36 genes that allows quantification of a complicated biological process such as angiogenesis, because it is known that expressions of each gene intertwine with each other during these processes and measurements of a single gene most likely is not enough to grasp the whole picture. In alignment with this notion, we found that the angiogenesis score associates with other known angiogenesis-associated gene expressions except for VEGFA, which happened to be the only gene that was included in the score among the ones we analyzed. Further, the intratumoral angiogenesis score was associated with immune reaction-, inflammation-, and metastasis-related pathways, and metastatic recurrence in breast cancer. This is in alignment with general expectation. However, the score did not correlate with survival, whereas pathologically determined angiogenesis is known to associate with breast cancer outcome [50,51]. One of the explanations for this difference may be the spatial location of whether angiogenesis is evaluated in intra- or peri-tumoral region of the bulk tumor. In TCGA, tumor samples used for transcriptome profiling were pathologically confirmed to have more than 60% of cells as cancer cells with no necrosis [52] that means viable center of the bulk tumor, whereas angiogenesis is commonly evaluated at the edge of the bulk tumor pathologically by microvessel density as the gold standard. The novel finding in this paper is that the angiogenesis score, although it is not a gold standard, allows assessment of angiogenesis in any sample with transcriptomic data.”

Comment 2:

In addition, while gene expression analysis is quite practicable, angiogenesis related factors, e. g. HIF1alpha are mainly regulated on posttranscriptional level (including localization). It seems especially intreguing that high angiogenesis score tumors should be more hypoxic. The authors should comment on this in regard to the general usefullness vs caveats of the angiogenesis score.

Response 2:

We totally agree with the Reviewer’s comment that HIF1-α's activity is regulated by oxygen levels at the post-transcriptional level (protein stability) and not transcriptional level. However, hypoxia, besides affecting HIF1-α at protein level, can also increase HIF1-α gene transcription (Rachida et al, Molecular Biology of the Cell, 2007). Further, we believe this highlights the caveat of using the score. We now note this in the Results sub-section.

Result 2.1. section:

Hypoxia-related genes, Hypoxia-inducible factor 1-alpha (HIF1A) was elevated in high score tumors in both cohorts. HIF1B (gene name: ARNT) was increased in high score tumors in the TCGA cohort, whereas it was increased in low score tumors in METABRIC cohort (Figure. 1D). By the nature of our study, the causality whether angiogenesis is the cause of hypoxia or vice versa is unknown. While hypoxia regulate HIF1A/HIF1B activity at the post-transcription level, it also increase HIF1A/HIF1B gene expressions [37]. This also highlights the caveat of using the score that although it does demonstrate association, the causality remains unknown. “

Comment 3:

This reviewer agrees with the authors, that a main point of tumor assessment is the site within the tumor where samples where taken. Especially ‚inside‘ the tumor necrosis could be expected. The authors should comment on this – maybe this was corrected for.

Response 3:

The samples analyzed in TCGA and METABRIC studies were collected under very strict criteria: >60% epithelial (cancer) cell content with no necrosis confirmed by pathology [49]. In our data analyses, we did not adjust comparisons for degree of necrosis because we did not have this information. We have re-written this part of Results to make this clearer. We have now added the sentences in the result 2.1 section as below,

Result section 2.1:

“Note that the samples obtained in these cohorts for gene expression measurement are representative of bulk tumors. Samples were confirmed to have high cancer cell content and no necrosis. To this end, we state our findings to be “intra-tumoral angiogenesis”. The common gold standard for evaluating angiogenesis is by immunostaining for determining micro-vessel density in peritumoral regions [35].”

Discussion

“Further, the intratumoral angiogenesis score was associated with immune reaction-, inflammation-, and metastasis-related pathways, and metastatic recurrence in breast cancer. This is in alignment with general expectation. However, the score did not correlate with survival, whereas pathologically determined angiogenesis is known to associate with breast cancer outcome [50,51]. One of the explanations for this difference may be the spatial location of whether angiogenesis is evaluated in intra- or peri-tumoral region of the bulk tumor. In TCGA, tumor samples used for transcriptome profiling were pathologically confirmed to have more than 60% of cells as cancer cells with no necrosis [52] that means viable center of the bulk tumor, whereas angiogenesis is commonly evaluated at the edge of the bulk tumor pathologically by microvessel density as the gold standard. The novel finding in this paper is that the angiogenesis score, although it is not a gold standard, allows assessment of angiogenesis in any sample with transcriptomic data.”

Comment 4:

The reference list should be more balanced, since more than half of the citations seem to come from the authors groups – here condensation would allow for citing pioneers of angiogenesis and of the key molecules involved.

Response 4:

We agree with the Reviewer that we should condense our publications and cite pioneers of angiogenesis and the key molecules involved. We removed Reference

#27 (Mukhopadhyay et al, Breast cancer management 2015, PMID: 27293484),

#33 (Nagahashi et al, BioMed research international 2014, PMID: 25133174),

#35 (Takabe et al, Gland Surgery 2012, PMID: 24855599),

#37 (Nagahashi et al, Advances in biological regulation 2014, PMID: 24210073),

#38 (Huang et al, Biomolecules 2013, PMID: 24286034),

and added references

#6 (Kottke et al, The Journal of clinical investigation 2010, PMID: 20364090),

#7 (Kim et al, The Journal of clinical investigation 2016, PMID: 27548529),

#8 (Korhonen et al, The Journal of clinical investigation 2016, PMID: 27548530),

#10 (Pyne et al, The Biochemical journal 2000, PMID: 10880336),

#15(Maceyka et al, Nature 2014, PMID: 24899305)

#16 (Obinata et al, International immunology 2019, PMID: 31049553),

#50 (Vermeulen et al, European journal of cancer 2002, PMID: 12142044),

#51 (Weidner et al, The New England journal of medicine 1991, PMID: 1701519),

which are landmark studies.

Minor points:

Comment 5:

Reference 4 does not define maximum tumor size without angiogenesis (it only states it).

Response 5:

We agree with the Reviewer that our original writing was inaccurate. We had modified the sentence in the Introduction section as below,

“It has been previously demonstrated that a tumor larger than 3mm in diameter was associated with angiogenesis [4].”

Comment 6:

Reference 24 is mainly lymphangiogenesis and does not refer to peritumoral localization of angiogenesis (measurements).

Response 6:

We agree with the Reviewer, and have now changed the reference to the correct paper below.

– “Miyata, Y et al. International Journal of Urology : official journal of the Japanese Urological Association 2015, 22, 806-815, doi:10.1111/iju.12840”.

Comment 7:

In Figure D the graph for HIF1B does not seem to display a higher value in median, yet the boxes of the plot are more condensed. This reviewer asked the authors to clarifiy ths impression of data presentation.

Response 7:

We agree with the Reviewer that the median values for the two groups are very similar. However, as seen with the 'condensed' boxes, on average, HIF1B expression (i.e., mean value) is significantly higher in the second group. The statistical test that is used to obtain the significant P value is based on mean and not median values. We consolidated the legend of X-axis of all the panels of Figure 1 to the bottom right corner, and make each panel larger. Further, the description below was added to Result section.

 Result 2.1. section:

Hypoxia-related genes, Hypoxia-inducible factor 1-alpha (HIF1A) was elevated in high score tumors in both cohorts. HIF1B (gene name: ARNT) was increased in high score tumors in the TCGA cohort, whereas it was increased in low score tumors in METABRIC cohort (Figure. 1D).”

Comment 8:

Are lines 125-131 on page 148 part of Figure Legend 1?

Response 8:

We agree with the Reviewer that the legend of the Figure 1 on the following page (lines 125-131) was confusing. It is a part of Figure 1 legend, and this is an error on layout, i.e. extra margin for the legend was not there. We have now fixed the extra margin for the legend for clarity.

Comment 9:

Some spelling and layout issues should be corrected.

Response 9:

We would like to thank the Reviewer for pointing out these issues. We have modified Figs. 1 and 4 so that their layout is clearer and easier to understand. We have also revised the entire manuscript to correct the layout and spelling mistakes.

Round 2

Reviewer 1 Report

In this second version of the paper there are many improvements.

The main problem is a huge misunderstanding on the hypothesis.

The authors should know that it is  a FAlSE idea to (still) think  that high density of angiogenesis, inside the tumor is a good sign for the efficacy of treatment. It is exactly the contrary. Angiogenesis in the tumor is pathologic and is the reason why it is useless and why hypoxia is not  reduced by the angiogenesis activity etc.

The series of demonstrated features which have shown that intratumor activity of angiogenesis is  a bad feature and not, what the authors expected (on the basis of one reference  from 2003 ref 41).

So to my opinion, the way of reasoning MUST be reconsidered and the conclusions will be positive (in fact expected).

The surprising data are in fact very expectable in view of the REAL intratumoral situation.  Please just think about it this way and the paper will take another meaning, much more interesting.

Besides this problem:

the methods and results are good and useful.

The English is (to this reviewer) very difficult to understand.

For example: what are the following sentences (although they have been corrected) meaning?

line 62 "thought to coincide to promote induce"

line 65 :" that could modulate both pro and anti-angiogenic properties [22-24], which is directly related..."

line 68 : "tumor’s ability to vessel invasion invade vessels and
metastasis to other areas of the body"

The following sentence is a false, old interpretation which totally  impairs the results of the paper:

line 155 : "score (Figure. 2A). The score was not associated with subtypes in  of the cohorts.
It has been proposed that tumors with enhanced angiogenesis respond to drug treatment better due to increased drug delivery with improved microcirculation in the tumor microenvironment [41].To this end, we examined whether angiogenesis score is associated with response to NAC using three independent cohorts, GSE21094 (paclitaxel/5-fluorouracil/cyclophosphamide and doxorubicin) GSE25066 (taxane and anthracycline), and GSE32646 (5/fluorouracil/epirubicin/cyclophosphamide and paclitaxel). Angiogenesis...."

Author Response

Author’s Point-by-point Response to Reviewer #1 Comments

preclinical studies [ijms-884943]

Manuscript: Intra-tumoral angiogenesis is associated with inflammation, epithelial mesenchymal transition and metastatic recurrence in breast cancer

Reviewer #1 (Remarks to the Author):

In this second version of the paper there are many improvements.

Response:

We would like to thank the Reviewer #1 for the encouraging remark, and for taking her/his time and effort reviewing our manuscript.

Comment 1:

The main problem is a huge misunderstanding on the hypothesis. The authors should know that it is a FALSE idea to (still) think that high density of angiogenesis, inside the tumor is a good sign for the efficacy of treatment. It is exactly the contrary. Angiogenesis in the tumor is pathologic and is the reason why it is useless and why hypoxia is not reduced by the angiogenesis activity etc... The series of demonstrated features which have shown that intratumor activity of angiogenesis is a bad feature and not, what the authors expected (on the basis of one reference from 2003 ref 41). So to my opinion, the way of reasoning MUST be reconsidered and the conclusions will be positive (in fact expected). The surprising data are in fact very expectable in view of the REAL intratumoral situation.  Please just think about it this way and the paper will take another meaning, much more interesting...

The following sentence is a false, old interpretation which totally impairs the results of the paper: line 155: "score (Figure. 2A). The score was not associated with subtypes in of the cohorts. It has been proposed that tumors with enhanced angiogenesis respond to drug treatment better due to increased drug delivery with improved microcirculation in the tumor microenvironment [41].To this end, we examined whether angiogenesis score is associated with response to NAC using three independent cohorts, GSE21094 (paclitaxel/5-fluorouracil/cyclophosphamide and doxorubicin) GSE25066 (taxane and anthracycline), and GSE32646 (5/fluorouracil/epirubicin/cyclophosphamide and paclitaxel). Angiogenesis...."

Response 1:

We agree with the Reviewer that the notion that high density of angiogenesis inside the tumor is a good sign for the efficacy of treatment is out of date, and that angiogenesis in the tumor is pathologic. We followed the Reviewer’s suggestion and have re-stated our reasoning for Figure 2.  Regarding the association of the angiogenesis score with response to chemotherapy, as per suggestion of the Reviewer, we have now corrected the Result 2.2. section as below.

Result 2.2:

“The score was not associated with subtypes in neither of the cohorts.

In the past, there was a notion that a high density of angiogenesis in a tumor is a good predictive indicator of treatment efficacy since improved microcirculation facilitates drug delivery to cancer cells [43]. However, it is now believed that angiogenesis in tumor can be pathologic and dysfunctional and does not improve microcirculation or drug delivery [44]. This aligns with our observation that the angiogenesis score does not negatively correlate with hypoxia (Figure 1D). To this end, it was of interest to examine whether the angiogenesis score is associated with response to NAC. For this, we utilized clinical and tumor gene expression data from three independent cohorts, with Gene Expression Omnibus repository identifiers GSE21094 (treated with paclitaxel/5-fluorouracil/cyclophosphamide and doxorubicin)[45], GSE25066 (taxane and anthracycline)[46], and GSE32646 (5/fluorouracil/epirubicin/cyclophosphamide and paclitaxel) [47]. As expected, the pre-NAC tumor angiogenesis score was not associated with the rate of pathological complete response (pCR) in patients with either estrogen receptor (ER)-positive/human epidermal receptor 2 (HER2)-negative or triple negative breast cancer (TNBC) tumors (Figure 2B).”

Comment 2:

the methods and results are good and useful.

Response 2:

We thank the Reviewer for appreciating the quality of our work.

Comments 3:

The English is (to this reviewer) very difficult to understand.

For example: what are the following sentences (although they have been corrected) meaning?

line 62 "thought to coincide to promote induce"

line 65 :" that could modulate both pro and anti-angiogenic properties [22-24], which is directly related..."

line 68 : "tumor’s ability to vessel invasion invade vessels and metastasis to other areas of the body"

Response 3:

We would like to thank the Reviewer for pointing out these issues. We have revised the entire manuscript to correct these and other English-related issues.

Line 62 > A multitude of conditions, including hypoxia and localized inflammation, coincide to promote tumor angiogenesis.

Line 65 > Pro-inflammatory cytokines such as interleukin (IL)-1β, IL-1α and IL-6 work through a variety of mediators, including tumor necrosis factor (TNF)α, to either enhance or suppress angiogenesis [22-24] to a degree that is directly related to cytokine concentration and duration of exposure [25,26].

Line 68 > A combination of these factors may contribute to the tumor’s ability to invade vessels and metastasize to other parts of the body [27-29].

Reviewer 2 Report

This reviewer thanks the authors for the detailed response to the comments.

The manuscript has been notably improved and several questions have been answered. Yet some issues remain:

Comment 1: While the genes set used are much better described, the issue of VEGFA still is not satisfactory explained.

Comment 5: The sentence in question has been rewritten – so verbatim it is now correct, yet the underlaying question of the origin of a 3 mm tumor size limit for angiogenesis is still not adressed.

Comment 6: The new citation relates now to angiogenesis, yet it is pancreatic tumor.

Comment 7: The authors explain in their response why they talk about a difference between the groups in the text (mean), yet in the box plots in Figure 1 is still only the median (which the authors conceed does not support their statement in the text. This needs to be clarified in the manuscript.

Author Response

Author’s Point-by-point Response to Reviewer #2 Comments

preclinical studies [ijms-884943]

Manuscript: Intra-tumoral angiogenesis is associated with inflammation, epithelial mesenchymal transition and metastatic recurrence in breast cancer

Reviewer #2 (Remarks to the Author):

This reviewer thanks the authors for the detailed response to the comments. The manuscript has been notably improved and several questions have been answered. Yet some issues remain:

Response: We would like to thank Reviewer #2 for taking his/her time and effort to review our manuscript, and for his/her comments that clearly improved it. 

Comment 1:

Using the angiogenesis score (a set of gene expression) the authors show similarities to known angiogenesis related genes namely the VEGF group, yet get non-expected results for clinical outcomes and immune infiltration. Only the correlation of angiogenesis score to metastasis meet the general expectations of high angiogenesis being favorable for tumor growth and metastasis. In addition the authors state (and list in Table S1) that only on gene of the VEGF group – and the only gene not reacting in the ‚correct‘ direction is included in the angiogenesis score set. This needs to be explained and discussed in much more detail, since it seems to this reviewer, that the genes set for angiogenesis score needs improvement.

à Comment 1: While the genes set used are much better described, the issue of VEGFA still is not satisfactory explained.

Response 1:

We agree with the Reviewer that the issue of VEGFA should be explained and discussed in more detail. We have re-written the Result 2.1 section as below.

Result 2.1 section:

“VEGFs are well-studied angiogenetic factors. As shown in Figure 1A, the expression of multiple VEGF-related genes (VEGFA, VEGFB, VEGFR1 (also known as FLT1), VEGFR2 (KDR), and VEGFR3 (FLT4), were significantly increased in high angiogenesis score-tumors. However, the angiogenesis score had no association with expression of VEGFA, the only gene among those examined for Figure 1 that is in the Hallmark angiogenesis gene set (Table S1). The pathway score thus significantly correlated with expression levels of four of the five angiogenesis-related genes, suggesting that the score adequately delineates intra-tumoral angiogenesis. Our angiogenesis score calculation considers the expression of 35 other genes besides VEGFA. This allows quantification of a complicated biological process such as angiogenesis, because it is known that expressions of each gene intertwine with each other during these processes and measurements of a single gene most likely is not enough to grasp the whole picture. It therefore is possible, as seen in Figure 1, for expression of VEGFA to not correlate with the angiogenesis score.”

Comment 2:

Reference 4 does not define maximum tumor size without angiogenesis (it only states it).

à Comment 2: The sentence in question has been rewritten – so verbatim it is now correct, yet the underlaying question of the origin of a 3 mm tumor size limit for angiogenesis is still not adressed.

Response 2:

We agree that the statement on the 3-mm tumor-size lacks strong support. We have therefore removed this statement.

Comment 3:

Reference 24 is mainly lymphangiogenesis and does not refer to peritumoral localization of angiogenesis (measurements).

àComment 3: The new citation relates now to angiogenesis, yet it is pancreatic tumor.

Response 3:

We would like to thank the Reviewer for pointing out our oversight. We agree with the Reviewer and have now changed the reference to the correct paper below.

Reference #36: “Shrivastav et al. Clinical breast cancer 2016”

Comment 4:

In Figure D the graph for HIF1B does not seem to display a higher value in median, yet the boxes of the plot are more condensed. This reviewer asked the authors to clarifiy ths impression of data presentation.

àComment 4: The authors explain in their response why they talk about a difference between the groups in the text (mean), yet in the box plots in Figure 1 is still only the median (which the authors conceed does not support their statement in the text. This needs to be clarified in the manuscript.

Response 4: In Figure 1, HIF1B gene expression is shown for two cohorts, TCGA and METABRIC. HIF1B expression is increased in high-score tumors of only TCGA, but not METABRIC cohort. We have clarified this in the text as below.

Result 2.1 section:

“The hypoxia-related gene, Hypoxia-inducible factor 1-alpha (HIF1A) was elevated in high score tumors in both cohorts. Another hypoxia-related gene, HIF1-beta (HIF1B, or ARNT) was increased in high score tumors in the TCGA cohort, but not in the METABRIC cohort (Figure 1D).”

“This allows quantification of a complicated biological process such as angiogenesis, because it is known that expressions of each gene intertwine with each other during these processes and measurements of a single gene most likely is not enough to grasp the whole picture.”

Round 3

Reviewer 1 Report

The manuscripot has been improved. It is possible ( to my opinion ) to publish it now except that it would be good if the authors would consider  in their analysis some of the key pathways as i asked for in my previous review like Notch4 and  tumor suppressors as p53, PTEN, VHL. It is indeed difficult to talk about angiogenesis if those molecules are not taken into account.

Reviewer 2 Report

This reviewer thanks the authors for adressing the comments.

Author Response

Reviewer #2 (Remarks to the Author):

This reviewer thanks the authors for adressing the comments.

Response:

We would like to thank Reviewer #2 for taking his/her time and effort to review our manuscript, and for his/her comments that clearly improved it.